# Hepatitis A Outbreak in Men Who Have Sex with Men Using Pre-Exposure Prophylaxis and People Living with HIV in Croatia, January to October 2022

**DOI:** 10.3390/v15010087

**Published:** 2022-12-28

**Authors:** Nikolina Bogdanić, Josip Begovac, Loris Močibob, Šime Zekan, Ivana Grgić, Josip Ujević, Oktavija Đaković Rode, Snježana Židovec-Lepej

**Affiliations:** 1University Hospital for Infectious Diseases, 10 000 Zagreb, Croatia; 2School of Medicine, University of Zagreb, 10 000 Zagreb, Croatia; 3School of Dental Medicine, University of Zagreb, 10 000 Zagreb, Croatia

**Keywords:** hepatitis A, men who have sex with men, epidemic, pre-exposure prophylaxis, HIV

## Abstract

The hepatitis A virus (HAV) is a highly hepatotropic virus transmitted mainly via the fecal–oral route. The purpose of this study is to describe a prolonged HAV outbreak in HIV-infected men who have sex with men (MSM) and pre-exposure prophylaxis (PrEP) users in Croatia in 2022. Croatia has a centralized system of HIV care and the PrEP service is only available at the University Hospital for Infectious Diseases (UHID), Zagreb. We reviewed all MSM living with HIV and MSM PrEP users at UHID and identified those diagnosed with HAV between January and October 2022. During this period, a total of 1036 MSM living with HIV and 361 PrEP users were followed, and 45 (4.4%) and 32 (8.9%) were diagnosed with HAV, respectively. Most cases were diagnosed in mid-February. A total of 70.1% (726/1036) MSM living with HIV and 82.3% (297/361) PrEP users were susceptible to HAV. Sequencing information was available for 34 persons; in all cases the HAV subtype IA was found. Our findings indicate that both MSM living with HIV and HIV-uninfected PrEP users are vulnerable to HAV infection and might be a potential source for a more widespread HAV epidemic.

## 1. Introduction

The hepatitis A virus (HAV) is a small, non-enveloped noncytopathic and highly hepatotropic virus from the genus *Hepatovirus* within the family Picornaviridae, transmitted via the faecal–oral route through direct contact and through contaminated food or water. In 2016 and before, European Union and European Economic Area (EU/EEA) countries have had a low incidence of HAV infection with most cases reported in children with slightly higher notification rates among males than females [1]. Hepatitis A has been also associated with travel to endemic countries or consumption of contaminated imported food. In Croatia, HAV vaccination is generally recommended to travelers to endemic regions (such as Latin America, Africa and Asia) and as post-exposure prophylaxis (within two weeks of exposure) during an outbreak [2]. However, availability of the vaccine is limited.

A large and prolonged epidemic in men who have sex with men (MSM) was noticed in many EU/EEA countries starting in 2016 and lasting until 2018 [3,4]. In 2017, a total of 19,947 HAV cases were reported by 24 EU/EEA countries, which represented more than a fourfold increase compared with 2012–2015. It was associated with MSM and peaked in March 2017 when the male to female ratio was 11.8. [3]. It involved one of the three HAV genotype IA outbreak strains: VRD_521_2016, RIVM-HAV16-090 (EuroPride), and V16-25801 (UK/Spain) [1,3,4]. The epidemic strains kept circulating two years from the first detection and were detected more often in men than women [3].

The aim of this study is to describe an ongoing hepatitis A outbreak among HIV-infected MSM and MSM who use pre-exposure prophylaxis (PrEP) in Croatia that started in 2022. We were interested in describing the outbreak in the context of immunity to HAV and also report the main molecular findings of the HAV in the current epidemic. A brief overview of the HAV outbreaks among MSM in Europe from 2016 to 2021 is also given.

## 2. Materials and Methods

### 2.1. Setting

Croatia has a centralized system of HIV care, and all persons living with HIV (PLWH) are treated at the University Hospital for Infectious Diseases (UHID) in Zagreb [5,6,7]. The PrEP service was introduced in Croatia as a pilot project in September 2018, and this service is still available only at the HIV Outpatient Department at the UHID in Zagreb. The HAV vaccine has never been administrated to PLWH and PrEP users on-site at UHID.

### 2.2. Participants

We reviewed all PLWH and PrEP users who used the service at UHID and identified those who were diagnosed with hepatitis A between January and October 2022. Included in the analysis were MSM ≥ 18 years old. The diagnosis of HAV was established upon the detection of serum HAV-specific IgM antibodies (LIAISON^®^ HAV IgM, DiaSorin, Saluggia, Italy). We also reviewed the HAV antibody status and assessed whether evidence of past syphilis was present by a positive *Treponema pallidum* hemagglutination test (Newbio-TPHA, NewMarket Biomedical Ltd., Kentford, UK). If antibodies to HAV (LIAISON^®^ anti-HAV, DiaSorin, Saluggia, Italy) were present before 2022, the person was considered to be immune to hepatitis A. 

If antibodies to HAV were not present in a blood sample analyzed in 2022 or if there was evidence of seroconversion in 2022, the person was considered susceptible to hepatitis A. Data on age and residence were also retrieved from person’s records. 

### 2.3. Hepatitis A Virus Genotyping

HAV genotyping was performed in 34 patients with detectable HAV RNA from the 2022 epidemics. Nucleic acid extraction was performed by using QIAmp Viral RNA Mini Kit (Qiagen, Hilden, Germany). Detection of HAV RNA was performed by using Altostar HAV RT-PCR kit (Altona diagnostics, Hamburg, Germany) on LightCycler 480 instrument (Roche Diagnostics, Rotkreuz, Switzerland). 

For HAV RNA sequencing, RT-PCR was performed by using “SuperScript^TM^ III One-Step RT-PCR System with Platinum^®^ Taq High Fidelity” (Invitrogen, Carlsbad, CA, USA, SAD) and primersVP1f1 5′GTTTTGCTCCTCTTTATCATGCTATGU3′ VP1r1 at 50 °C for 30 min, 94 °C for 2 min, 45× (94 °C for 30 s, 45 °C for 30 s, 68 °C for 90 s) and 68 °C for 5 min. Nested PCR was carried out with the “FastStart™ High Fidelity PCR System” (Roche Diagnostics, Mannheim, Germany), primers VP1Nf2 5′TATCATGCTATGGATGTTAC3′ and VP1Nr2 5′TTCATTATTTCATGCTCCTC3′ at 94 °C for 2 min, 45× (94 °C for 30 s, 60 °C for 30 s, 72 °C for 60 s) and 72 °C for 7 min. Sequencing reaction was performed by using BigDye^TM^ Terminator v3.1 Cycle Sequencing Kit (Applied Biosystems, Warrington, UK) with primers used for nested PCR. The reaction was carried out at 25× (96 °C for 10 s, 50 °C for 5 s, and 60 °C for 4 min). The sequences were analysed in the “Vector NTI” program, and the genotype and subtype of VP1-2A HAV genome region was determined by using the Hepatitis A Virus Genotyping Tool Version 1.0 algorithm (https://www.rivm.nl/mpf/typingtool/hav/#:~:text=This%20tool%20is%20dsigned%20to,20000%20squences%20at%20a%20time, access date 5 November 2022). CustalW within MEGA X was used for the alignment of the 45 obtained sequences, along with 12 sequences from the NCBI gene bank where 878 nucleotides was the final length of the sequences. The phylogenetic tree was constructed in the MEGA X program using the maximum likelihood method, Tamura-Nei evolutionary model with 1000 bootstrap replicates. Sequences of the VP1-2A region from the 2022 epidemics are available in the GenBank under accession numbers OP807977-OP808012. Phylogenetic analysis also included 35 sequences from the Croatian HAV epidemics in the years 2017 and 2018 available in the GenBank under accession number MK396845-MK396889.

### 2.4. Statistical Analysis

We describe our findings with the median and first and third quartiles for continuous variables and by frequencies for categorical variables. Continuous variables were compared using the Mann–Whitney test; categorical variables were compared using the Chi-square test. The trend in the susceptibility to hepatitis A by age categories was assessed using the Cochrane–Armitage test. 

We used GraphPad Prism version 9.4 for Windows (GraphPad Software, San Diego, CA USA) to make the graphs and SAS software 9.4 (SAS Institute, Cary, NC, USA) to perform the analyses.

### 2.5. Review of HAV Epidemics in MSM in EU/EEA

We searched PubMed for research articles published up to October 31, 2022, with no language restrictions, using the following sets of search terms: “HAV”, “MSM” and “Europe”. In addition, European Centre for Disease Prevention and Control epidemiological reports and surveillance data were included. Studies conducted in Europe with abstracts published in English were included. Studies not conducted in Europe, and studies that did not include any relevant data on epidemic characteristics and included only serology, molecular, vaccination, behavior or epidemiological data were excluded from the review. Finally, 28 studies were included and described in the discussion section.

## 3. Results

A total of 1036 PLWH and 361 PrEP users were followed during the period from 1 January 2022 to 1 October 2022 (Table 1). During this period, almost all PLWH were using ART, and among those with an available viral load measurement in 2022, 95% had <50 copies of plasma HIV-RNA per milliliter. Of 1036 PLWH, 70.1% and 82.3% of PrEP users were susceptible to HAV (Table 1). A total of 45 PLWH and 32 PrEP users diagnosed with hepatitis A were enrolled in the study (Table 2). The majority of HAV cases were identified in February (Figure 1). Both PLWH and PrEP users were young. The majority were 25–44 years old, but PrEP users were somewhat younger than PLWH (median age, 35.8 vs. 39.4 years old, respectively, *p* = 0.087). The median duration of HIV infection was 5.5 years, and more than 90% were using antiretroviral treatment (ART). Of 45 PLWH, 41 had a VL < 50 copies/mL. Of four participants with plasma HIV-RNA > 50 copies/mL, two had a concomitant HIV and hepatitis A diagnosis, one had interrupted antiretroviral therapy, and one had only 1-month of ART. Among persons with hepatitis A, more than ¾ of PLWH and less than half of PrEP users had syphilis in the past (*p* = 0.002, Table 2). About ¾ of PLWH and PrEP users diagnosed with HAV were living in Zagreb. Susceptibility to HAV was highest among both PLWH and PrEP users 18–30 years old and decreased with age (*p* < 0.001 for the trend test for both PLWH and PrEP users, Figure 2).

MSM, men who have sex with men; PLWH, people living with HIV.

### Genotype Analysis

All sequences of the HAV VP2/P2A region obtained during the 2022 epidemics as well as sequences from the 2017/2018 epidemics were determined to be subtype IA. Phylogenetic analysis of sequences showed three distinct clusters: sequences from the 2022 epidemics (n = 34), RIVM-HAV-090 strain cluster from the 2018 epidemic (n = 18, EuroPride strain) and VRD_521_2016 strain cluster from the 2017/2018 epidemic (n = 17, UK/Spain) (Figure 3).

## 4. Discussion

We report an increase in the number of persons diagnosed with hepatitis A among MSM living with HIV and MSM PrEP users in Croatia from January to October 2022. Most cases were diagnosed in mid-February (Figure 1). During the period from January 2022 to October 2022, a total of 1036 PLWH and 361 PrEP users were followed, and 45 (4.4%) and 32 (8.9%) were diagnosed with HAV, respectively. 

Hepatitis A outbreaks were described among PLWH and PrEP users during the 2017/2018 European epidemic [8,9,10,11,12,13]. In a report from two tertiary centers in northern Italy during the HAV outbreak, MSM status and HIV infection did not have an impact on the clinical course nor severity of illness [14]. In Croatia, there was not a statistically significant difference in age among PLWH and PrEP users (39.4 vs. 35.8 years old), while in Lyon, France PLWH were significantly older than PrEP users (49 vs. 36 years old, *p* < 0.001) [10]. 

In southwest Hungary during the period 2010–2020, the HAV seropositivity was low (less than 18%) up to the age of 40 years and increased to 75–80% at age group 66–70 years [15]. Similarly, susceptibility to HAV in our study was highest in younger PLWH and PrEP users and decreased with age (Figure 2). In our study, HAV seropositivity in MSM PrEP users was 16.9% which is between figures reported from two studies conducted in HIV uninfected MSM in Croatia which found a prevalence of 20.4% in 2011 and of 14.2% in 2006 [16]. Due to fecal–oral transmission during sexual practices, the risk of HAV is particularly high among MSM, and outbreaks may be a result of a low level of immunity. However, outbreaks may occur despite a high level of immunity. For example, among 2023 PLWH and 415 PrEP users followed in Lyon, France, from January to June 2017, 60.3% of PLWH and 73.5% of PrEP users were immune to HAV (20.6% and 38.1% due to vaccination, respectively) [10]. In a report from the UK, of 181 HIV-negative MSM who received sexual health services from March to August 2017, 54.5% were susceptible to HAV [17]. During the HAV outbreak in Berlin, vaccination coverage among 756 MSM was 32.7% before 2017 [18]. As there is a recommendation for the HAV vaccine for all MSM in France, there was no significant difference between PLWH and PrEP users in HAV-susceptible MSM in a study from Lyon (PLWH: 26.6%, PrEP users: 24.9%, *p* = 0.48) [10]. In our study, the proportion of HAV-immune persons was higher among PLWH than PrEP users (29.0 vs. 16.9%, respectively, *p* < 0.001). This could be partially explained by the older age of PLWH and possibly by some behavioral factors and willingness to receive the vaccine. However, the HAV vaccine in Croatia is recommended only for travelers and as post-exposure prophylaxis, but due to limited amounts and frequent shortages, it was not possible to provide the vaccine for all MSM at risk at UHID in the current epidemic.

Previous studies showed that patients diagnosed with HAV are often coinfected with another sexually transmitted disease (STD) [19,20]. Almost half of the patients diagnosed with HAV in northern France reported high risk sexual behaviors and a history of STD [9]. Of the PLWH and PrEP users followed at UHID and diagnosed with HAV, 77.8% and 43.8%, respectively, previously had syphilis, which is an indication of high-risk sexual behaviors. 

After an increase of confirmed HAV cases in 2017 in EU/EEA countries, including Croatia, which was associated with younger MSM [3], the number of HAV cases dropped to pre-epidemic levels in the following years (Figure 4) (https://www.ecdc.europa.eu/en/hepatitis-a/surveillance/atlas, access date 5 November 2022). After 96 HAV cases were reported in Croatia in 2018, there were only 9, 5 and 5 cases reported in 2019, 2020 and 2021, respectively (https://www.ecdc.europa.eu/en/hepatitis-a/surveillance/atlas). However, there is a possibility of under-reporting, and these numbers might underestimate actual case numbers.

Our report on the HIV outbreak has some limitations. We included only patients diagnosed with HAV followed at the HIV Outpatient Department at UHID. This could result in underestimation of case numbers as some persons might seek help elsewhere. In addition, we did not have any data on recent travel or other risk behaviors.

### 4.1. Brief Overview of the European 2017/2018 Outbreak

The European outbreak of hepatitis A during 2017/2018 was reported from a number of countries. The increase in HAV cases was observed mostly among younger MSM (Table 3) [4,8,9,10,11,12,13,14,19,20,21,22,23,24,25,26,27,28,29,30,31,32,33,34]. In a study from the Netherlands among MSM visiting the Sexually Transmitted Infection Clinic in Amsterdam, HAV seropositivity determinants in a multivariable analysis were older (*p* < 0.001), came from an HAV endemic country (*p* < 0.001) and were HBV seropositive (*p* = 0.001) [35]. During November 2016 to December 2017 in Asturias, Spain, the HAV outbreak was associated with more severe hepatitis and more cases with positive syphilis serology compared to the period from January 2004 to September 2016 [22]. In Vienna, there was an increase of severe HAV infections among young males in 2016/2017: 21.5% vs. 8.0% in 2008–2016 with liver dysfunction, *p* < 0.001 [36]. Between January and June 2017, 44 HAV cases were registered at the hospital in Barcelona; the vast majority (96%) were male, 67% were MSM, and 39% required hospitalization [37]. Table 3 gives an overview of studies on the 2016/2017 European HAV epidemic among MSM published through September 2022.

The response to an HAV outbreak among MSM should include expanded vaccination accessibility as well as involvement and good communication with the community, including using social media. For example, in Normandy, France, effective vaccination campaigns included HAV vaccination sexually transmitted infection clinics, at community-based sites and on mobile trucks. Vaccination and sexual health promotion through social media and gay dating apps have also been implemented [25]. Because of a limited vaccine supply, a widespread vaccination campaign for MSM was not possible in Croatia.

### 4.2. An Overview of Molecular Epidemiology of Hepatitis A Epidemics in Europe from 2016 to 2022

In October and December 2016, the Netherlands, followed by the United Kingdom, reported two clusters of hepatitis A virus sub-genotype IA infections. Later, the investigation of these cases uncovered an HAV outbreak disproportionally affecting MSM [3]. During 2016, an outbreak of HAV infection was observed in the Malaga province in Spain among 51 patients. The majority were male (90%) with a mean age of 35.7 years, and 55% were MSM. Half of the patients were hospitalized due to significant coagulopathy; however, there were no cases of fulminant hepatic failure [38].

In 2017, a total of 19,947 HAV cases (mostly genotype IA) were reported by 24 EU/EEA countries which represented more than a fourfold increase compared with the average of 4671 cases reported in 2012–2015. The confirmed number of cases provides an underestimation of the true extent of the outbreak due to the lack of sequencing information. This outbreak was also associated with MSM [3]. There was an increase in the number of confirmed HAV cases in 2017, mostly in age group 25–44 years (Figure 4). Between June 2016 and May 2017, 17 EU/EEA countries reported 4096 HAV cases of which 1400 (34%) were confirmed. Genotypes IA, VRD_521_2016 and RIVM-HAV16-090 accounted for 92% of the cases. The epidemiological data were available for only 393 confirmed cases: 84% were MSM and 92% were unvaccinated [34]. In Latvia during 2017–2019, 179 laboratory-confirmed HAV cases were analyzed, and 69.8% were typed. The majority were genotype VRD_521_2016, followed by RIVM-HAV16-090 and V16-25801 [4]. Between July 2016 and January 2017, across England and Northern Ireland, 37 HAV cases were confirmed of which 28 identified as MSM, 24 were strain VRD_521_2016 with a clear relation to travelers returning from Central and South America, and 13 were strain RIVM-HAV16–090 related to EuroPride 2016 which was held in Amsterdam in July/August 2016 [39].

In the period July 2016 to February 2017, 48 HAV cases were confirmed in the Netherlands; 17 were MSM, and 10/13 cases with available typing information were infected with RIVM-HAV16-090 (EuroPride strain) [29].

Similar to the epidemiological situation in the abovementioned European countries, HAV strains RIVM-HAV16-090 and VRD_521_2016 were detected among MSM from Croatia in the period 2017-2018. HAV genotyping of IA sequences from the current Croatian epidemics among MSM revealed a cluster of IA genotype sequences that was closely related but not identical to the RIVM-HAV16-090 and VRD_521_2016 local strains from 2017–2018, suggesting a possibility of a distinct source of infection that needs to be a subject of additional epidemiological monitoring and characterization in the near future. 

## 5. Conclusions

We described a prolonged hepatitis A outbreak among HIV-infected MSM and PrEP users in Zagreb, Croatia, that started in January 2022. Our findings indicate that both MSM living with HIV and HIV-uninfected PrEP users in Croatia are vulnerable to HAV infection and might be a potential source for a more widespread HAV epidemic. To avoid future hepatitis A outbreaks, HAV vaccination for all MSM should be implemented. There is an opportunity to implement vaccination “on-site” at the HIV Outpatient Department and PrEP service at UHID, Zagreb, Croatia.

## Figures and Tables

**Figure 1 viruses-15-00087-f001:**
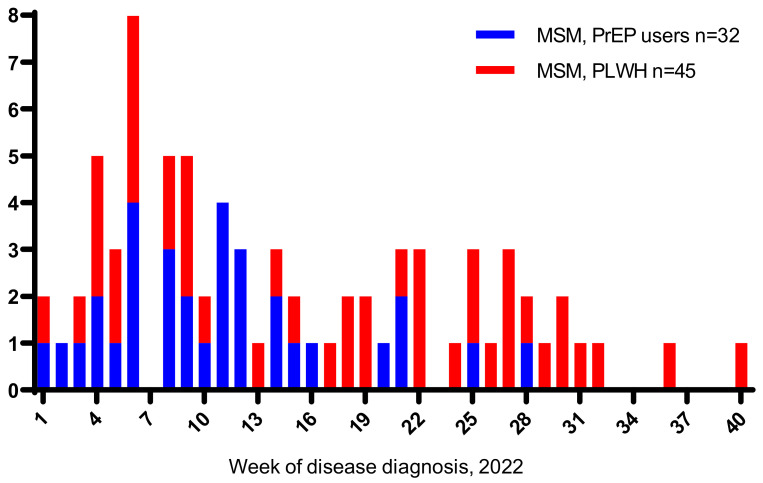
Epidemic curve of total hepatitis A cases among men who have sex with men (MSM) who use pre-exposure prophylaxis (PrEP) and MSM living with HIV (PLWH), January–October 2022 according to weeks, Croatia (n = 77).

**Figure 2 viruses-15-00087-f002:**
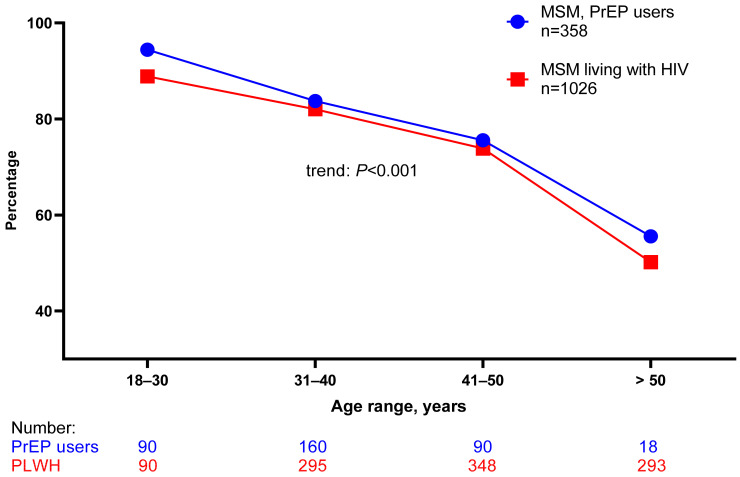
Proportion of individuals susceptible to hepatitis A virus infection according to age, January–October 2022, Croatia. Data for 10 MSM living with HIV and 3 HIV uninfected MSM were unknown. The *p*-value from the Cochrane–Armitage trend test was the same for both MSM PrEP users and MSM living with HIV.

**Figure 3 viruses-15-00087-f003:**
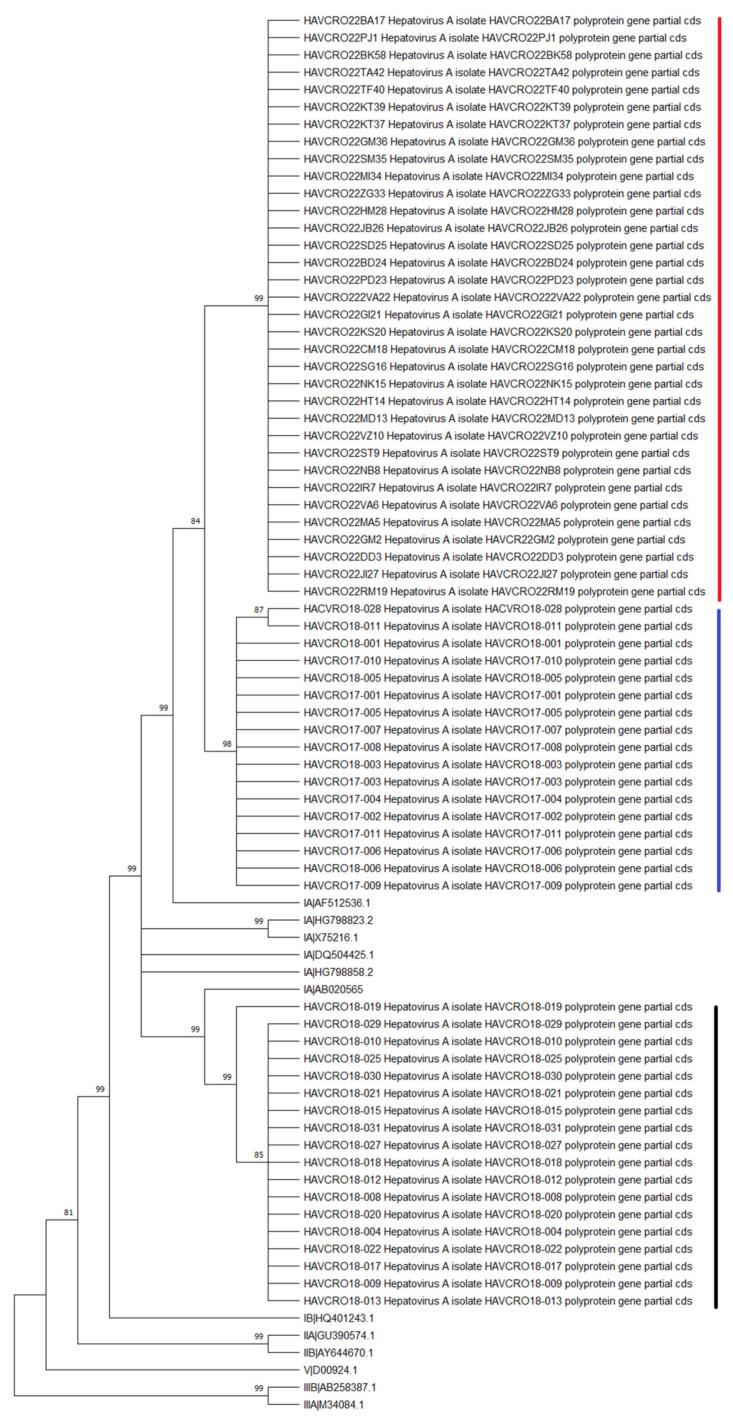
Phylogenetic tree of the VP1/P2A region of the HAV genome constructed using a maximum likelihood method, Tamura–Nei model showing three distinct clusters: Croatian2022 sequences (marked in red), RIVM-HAV-090 strain sequences (EuroPride, 2018, in black) and VRD_521_2016 sequences (UK/Spain, 2017–2018, blue).

**Figure 4 viruses-15-00087-f004:**
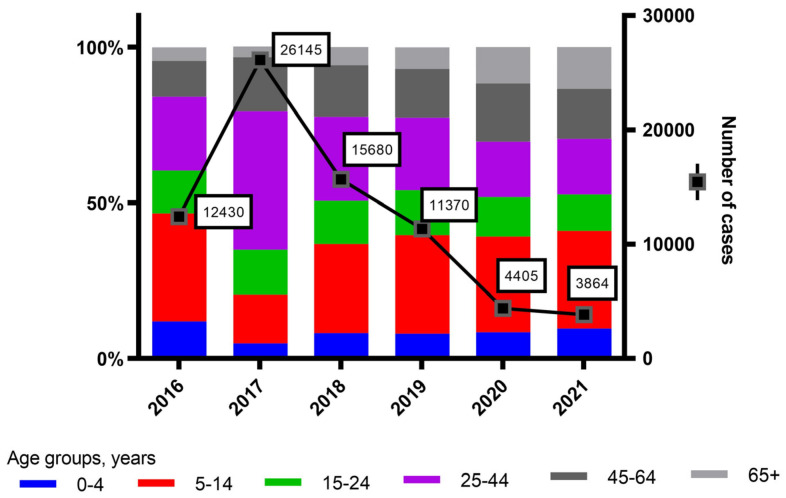
Number of cases of hepatitis A in EU/EEA countries by age groups, ECDC 2016 to 2021. Source: https://www.ecdc.europa.eu/en/hepatitis-a/surveillance/atlas, access date 5 November 2022.

**Table 1 viruses-15-00087-t001:** Characteristics of HIV-infected MSM (n =1036) and MSM PrEP users (n = 361) followed in Croatia, January–October 2022.

Characteristics	HIV-Infected MSM	MSM PrEP Users
Age, median (Q1–Q3)	44.6 (37.5–52.2)	35.9 (30.1–42.2)
CD4 cells count per mm^3^ median (Q1–Q3)	668 (475.0–859.0) ^a^	NA
Antiretroviral treatment ^b^	1028 (99.4)	NA
HIV viral load < 50 copies/mL, %	842 (94.9) ^c^	NA
Hepatitis A status		
Previous infection or vaccinated ^d^	300 (29.0)	61 (16.9)
Susceptible	726 (70.1)	297 (82.3)
Unknown	10 (1.0)	3 (0.8)
Ever had syphilis ^e^	479 (46.2)	122 (33.8)
Living in Zagreb		
Yes	538 (51.9)	209 (57.9)
Unknown	7 (0.7)	27 (7.5)
Had hepatitis A during outbreak	45 (4.4)	32 (8.9)

^a^ Based on 734 persons with an available measurement in 2022; ^b^ collected at least one one-month supply of antiretroviral drugs during January–October 2022; ^c^ of 887 persons with an available viral load measurement in 2022; ^d^ had antibody to hepatitis A; ^e^ had a *Treponema pallidum* Hemagglutination Assay Serum Titer ≥ 80; MSM, men who have sex with men; PrEP, pre-exposure prophylaxis for HIV infection; NA, not applicable; values are frequencies, percentages or median and first and third quartiles.

**Table 2 viruses-15-00087-t002:** Characteristics of HIV-infected MSM (n = 45) and MSM PrEP users (n = 32) diagnosed with hepatitis A, Croatia, January–October 2022.

Characteristics	HIV-Infected MSM N = 45	MSM Prep Users N = 32
Age, years	39.4 (33.3–45.6)	35.8 (29.6–42.1)
Age groups		
≤24 years	2 (4.4)	2 (6.3)
25–44 years	31 (68.9)	25 (78.1)
44–64 years	12 (26.7)	5 (15.6)
CD4 cells count per mm^3^	778.0 (590.0–1015.0)	NA
Antiretroviral treatment	42 (93.3)	NA
HIV viral load < 50 copies/mL	41 (91.1)	NA
Known duration of HIV infection, years	5.5 (3.2–9.3)	NA
Had syphilis in the past ^a^	35 (77.8)	14 (43.8)
Living in Zagreb	33 (73.3)	24 (75)
Hepatitis A genotype N = (34)		
Subtype IA	19	15

^a^ Had a positive *Treponema pallidum* hemagglutination titer of ≥1:80; NA, not applicable; MSM, men who have sex with men; PrEP, pre-exposure prophylaxis; values are frequencies with percentages or median and first to third quartile.

**Table 3 viruses-15-00087-t003:** Studies of hepatitis A epidemics among men who have sex with men with known number of cases and basic epidemic characteristics in Europe from 2016 to 2021, published through September 2022 (n = 28).

Author	Place	Year of Outbreak	Number of HAV Cases	Epidemic Characteristics
García Ferreira AJ et al.	Malaga province, Spain	2016	51	90% male, 55% MSM
Penot P et al.	Paris, France	2016/2017	3	MSM (all PrEP users)
Beebeejaun K et al.	England and Northern Ireland	2016/2017	37	28/37 MSM
Werber D et al.	Berlin, Germany	2016/2017	38	97% male, 81% MSM
Comelli A et al.	Brescia, Italy	2016/2017	42	60% MSM
Freidl GS et al.	Netherlands	2016/2017	48	17/48 MSM
Nicolay N et al.	Normandy, France	2016/2017	78 *	88% male, 69% MSM
Fraile M et al.	Asturias, Spain	2016/2017	108	95.4% male, 63.9% MSM
Ciccullo A et al.	Rome, Italy	2016/2017	141	92.9% male, 70.2% MSM
Lanini S et al.	Lazio, Italy	2016/2017	513	87.5% male, no data on MSM
Ndumbi et al.	EU/EEA countries	2016/2017	4096 **	565/676 MSM ***
Zimmermann et al.	Berlin, Germany	2016–2018	222	95.6% male, 84.7% MSM
Zucman D et al.	France	2017	5	MSM
Ormarsdottir S et al.	Iceland	2017	5	4 male, all MSM
Rossati A et al.	Novara, Italy	2017	14	79% male, all MSM, 3 PLWH, 2 new syphilis diagnosis
Friesema IH et al.	Netherlands	2017	374	293/374 male, 59% MSM
Martin A et al.	Marseille, France	2017	20	PLWH, 100% MSM (recently diagnosed with HIV and another STI)
Alventosa-Mateu C et al.	Valencia, Spain	2017	23	78% male, 67% MSM
Rodríguez-Tajes S et al.	Barcelona, Spain	2017	44	96% male, 67% MSM
Lombardi A et al.	Northern Italy	2017	117	91% male, 66% MSM
Aulicino G et al.	Milan, Italy	2017	353	172/353 MSM
Raczyńska A et al.	Krakow, Poland	2017/2018	119	88% male, 71% MSM
Charre C et al.	Lyon, France	2017	44	PLWH and PrEP users; 38/44 male, 33/38 MSM
Mauro MV et al.	Cosenza, Italy	2017	27	81% male, 64% MSM
Boucher A et al.	North of France	2017	49	98% male, 65% MSM, 11 PLWH
Bazzardi R et al.	North Sardinia, Italy	2017/2018	10	10 male, 8 MSM, 2 new HIV+
Dimitriou P et al.	Cyprus	2017/2018	13	100% male, 31% MSM
Savicka O et al.	Latvia	2017–2019	125	51% male, 6% MSM

HAV, hepatitis A; MSM, men who have sex with men; PLWH, people living with HIV; STI, sexually transmitted infection; PrEP, pre-exposure prophylaxis; * 48 confirmed outbreak cases and 30 possible outbreak cases; ** 1400 confirmed outbreak cases; *** available data for 676 cases.

## Data Availability

The data that support the findings of this study are available from the authors upon reasonable request.

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
