# Peer review of "Hepatitis A Outbreak in Men Who Have Sex with Men Using Pre-Exposure Prophylaxis and People Living with HIV in Croatia, January to October 2022"

_viruses, 2022, doi:10.3390/v15010087_

Round 1

Reviewer 1 Report

Dear Editor,

Thanks for inviting me to review manuscript viruses-2106329 entitled "Hepatitis A outbreak in persons living with HIV and men who have sex with men using pre-exposure prophylaxis, Croatia, January to October 2022".

The title is interesting and the methodology seems correct. There are some comments regarding the manuscript that should be addressed by the authors before being published.

Title:

- The title might be unclear for some readers about the study population. I think it is better to say "men who have sex with men using pre-exposure prophylaxis and people living with HIV" because PLWH are not PrEP users.

- Also, there is no need to mention the study period in the title.

Abstract:

"We included into the analysis MSM who were diagnosed with HAV at UHID between January and October 2022." This sentence does not seem correct since the authors have analyzed people with and without HAV.

The last sentence of abstract is not based on the results. HAV vaccination is not evaluated in this study.

Introduction:

This section is too short. The authors need to expand this section using the importance of this subject for the literature, HAV in PLWH and MSM with and without PrEP, some other recent epidemiological studies, etc.

Methods:

Some sentences in the "setting" section belong to the introduction (e.g. HAV vaccination and travelers, MSM, PLWH, etc.), these sentences also need reference. 

The statistical analysis section needs further details about the test used to evaluate the normality of distributions and the software which is used to analyze the data.

Results:

- Please include the P values in the tables 1 and 2 and figure 2.

Conclusion:

Some sentences/recommendations of this section are not based on the results.

Figure 1:

Please provide a label for the Y axis of figure 1.

Please mention the time period JAN-OCT in the X axis.

Figure 2:

In the figure 2, please mention how many people are within these age groups. Since the figure may not have space for this data, you can mention this proportion in the table 1.

Figure 3:

It would be better if the authors mention the "Tamura-Nei model" beside the red mark and also RIVM-HAV-090 for blue and VRD_521_2016 for black marks.

Regards,

Reviewer

Author Response

Response to reviewers

We are grateful to the reviewers for the comments on our paper. A point-by-point response to the Reviewers’ comments, which are repeated in italics, is given below.

Reviewer 1

° Dear Editor,
Thanks for inviting me to review manuscript viruses-2106329 entitled "Hepatitis A outbreak in persons living with HIV and men who have sex with men using pre-exposure prophylaxis, Croatia, January to October 2022". The title is interesting and the methodology seems correct. There are some comments regarding the manuscript that should be addressed by the authors before being published.

Title:
- The title might be unclear for some readers about the study population. I think it is better to say "men who have sex with men using pre-exposure prophylaxis and people living with HIV" because PLWH are not PrEP users.
- Also, there is no need to mention the study period in the title.

Thank you for this suggestion. We corrected the title in „Hepatitis A outbreak in men who have sex with men using pre-exposure prophylaxis and people living with HIV in Croatia, January to October 2022“. We kept the timelines as it is important to emphasise the new epidemics.

° Abstract:
"We included into the analysis MSM who were diagnosed with HAV at UHID between January and October 2022." This sentence does not seem correct since the authors have analyzed people with and without HAV.

We agree, this sentence is not precise. We corrected it in „We reviewed all PLWH and PrEP users at UHID and identified those diagnosed with HAV between January and October 2022.“

° The last sentence of abstract is not based on the results. HAV vaccination is not evaluated in this study.

We excluded this sentence from the abstract. HAV vaccination was not evaluated in this study, but we reviewed HAV antibody status and described the outbreak in the context of immunity.

° Introduction:

This section is too short. The authors need to expand this section using the importance of this subject for the literature, HAV in PLWH and MSM with and without PrEP, some other recent epidemiological studies, etc.

As we included this data in the Discussion section, we slightly expended Introduction section to avoid repeating.

° Methods:

Some sentences in the "setting" section belong to the introduction (e.g. HAV vaccination and travelers, MSM, PLWH, etc.), these sentences also need reference. 

We included the sentence about HAV vaccination in the introduction and added the reference. We left the part about vaccination of MSM and PLWH at our hospital in the methods section.

° The statistical analysis section needs further details about the test used to evaluate the normality of distributions and the software which is used to analyze the data.

We used the Mann-Whitney test to compare the age of HIV+ MSM with the age of MSM PreP users. Since papers on hepatitis A among MSM PrEP users and HIV+ MSM report the median age (see Freidl GS etal.; Beebeejaun K et al.;  Bauer D et al. ), we also wanted to report the median age of our study population. When reporting the median it is more appropriate to use the Mann-Whitney test. We could have also used the t-test (“normality tests” indicated that they could be used), but then it we would be more appropriate to report the mean. The t-test on our data would give a P value of 0.094 which is very similar to the P=0.087 from the Mann-Whitney test. So, nothing in the interpretation of the data changes, hence we prefer to report the P-value from the Mann-Whitney test. We do mention the statistical software we used.

° Results:

 Please include the P values in the tables 1 and 2 and figure 2.

We wish not to report the P-Values in our Tables for several reasons.  Both tables contain HIV specific variables (CD4 cell cunt, HIV-1 RNA, antiretroviral treatment) which obviously cannot be compared between HIV+ and HIV- people. Also, some of the comparisons would be difficult to interpret. For example, the proportion of those in care who had hepatitis A among PLWH and PreP users. What would be the meaning of the “statistically non-significant P”? Furthermore, our study is not hypothesis driven, it is mainly a descriptive study. There has also been considerable controversy concerning the use of P-values in the literature. There was even a proposal to eliminate statistical significance testing, backed by over 800 signatories (Aguinis H, Vassar M, Wayant C. BMJ Evidence-Based Medicine 2021;26:39–42.; Amrhein V, Greenland S, McShane B. Scientists rise up against statistical significance. Nature 2019;567:305–7.).  Because of above-mentioned reasons we wish to keep the number of P-value to a minimum and only in the text.

° Conclusion:

Some sentences/recommendations of this section are not based on the results.

 We assume that the Reviewer refers to our reccomendation of HAV vaccination for all MSM. It is correct that our analysis was not focused on HAV vaccination. We had a data for previously infected or vaccinated persons based on HAV antibody status. However, as majority of MSM, both in HIV-infected and HIV-uninfected group were susceptible to HAV (70.1% and 82.3%, respectively) we think that our suggestion for HAV vaccination with regard to avoiding future epidemics is reasonable.

° Figure 1:Please provide a label for the Y axis of figure 1.

° Figure 2:In the figure 2, please mention how many people are within these age groups. Since the figure may not have space for this data, you can mention this proportion in the table 1.

°Figure 3:It would be better if the authors mention the "Tamura-Nei model" beside the red mark and also RIVM-HAV-090 for blue and VRD_521_2016 for black marks.(the figure itself was not modified but the explanation below was changed to make the markings more clear)

Figures have been adapted.

Reviewer 2 Report

Congratulation; this is an excellent study. Just minor issue: what is EEA?

Author Response

Reviewer 2

Congratulation; this is an excellent study. Just minor issue: what is EEA?

Thank you for this comment. EEA is European Economic Area. We included this in the revised version of the manuscript.

We thank you for your evaluation of our manuscript and consideration for publication.